# A Global Cndp1-Knock-Out Selectively Increases Renal Carnosine and Anserine Concentrations in an Age- and Gender-Specific Manner in Mice

**DOI:** 10.3390/ijms21144887

**Published:** 2020-07-10

**Authors:** Tim Weigand, Florian Colbatzky, Tilman Pfeffer, Sven F. Garbade, Kristina Klingbeil, Florian Colbatzky, Michael Becker, Johanna Zemva, Ruben Bulkescher, Robin Schürfeld, Christian Thiel, Nadine Volk, David Reuss, Georg F. Hoffmann, Marc Freichel, Markus Hecker, Tanja Poth, Thomas Fleming, Gernot Poschet, Claus P. Schmitt, Verena Peters

**Affiliations:** 1Centre for Paediatric and Adolescent Medicine, University of Heidelberg, 69120 Heidelberg, Germany; tim_weigand@gmx.de (T.W.); colba.flo@web.de (F.C.); tilman.Pfeffer@med.uni-heidelberg.de (T.P.); Sven.Garbade@med.uni-heidelberg.de (S.F.G.); Kristina.Klingbeil@med.uni-heidelberg.de (K.K.); robin.schuerfeld@t-online.de (R.S.); christian.Thiel@med.uni-heidelberg.de (C.T.); Georg.Hoffmann@med.uni-heidelberg.de (G.F.H.); clauspeter.schmitt@med.uni-heidelberg.de (C.P.S.); 2Nonclinical Drug Safety Germany, Boehringer Ingelheim Pharma GmbH & Co. KG, 88397 Biberach, Germany; florian.colbatzky@boehringer-ingelheim.com; 3Drug Metabolism & Pharmacokinetics Germany, Boehringer Ingelheim Pharma GmbH & Co. KG, 88397 Biberach, Germany; Michael.Becker@boehringer-ingelheim.com; 4Internal Medicine I and Clinical Chemistry, University Hospital Heidelberg, 69120 Heidelberg, Germany; Johanna.Zemva@med.uni-heidelberg.de (J.Z.); ruben.Bulkescher@med.uni-heidelberg.de (R.B.); thomas.fleming@med.uni-heidelberg.de (T.F.); 5Tissue Bank of the National Center for Tumor Diseases (NCT), 69120 Heidelberg, Germany; Nadine.Volk@med.uni-heidelberg.de; 6Neuropathology, University of Heidelberg, 69120 Heidelberg, Germany; David.Reuss@med.uni-heidelberg.de; 7Institute of Pharmacology, University of Heidelberg, 69120 Heidelberg, Germany; marc.freichel@pharma.uni-heidelberg.de; 8Department of Cardiovascular Physiology, University of Heidelberg, 69120 Heidelberg, Germany; hecker@physiologie.uni-heidelberg.de; 9Center for Model System and Comparative Pathology (CMCP), Institute of Pathology, University Hospital Heidelberg, 69120 Heidelberg, Germany; tanja.poth@med.uni-heidelberg.de; 10Centre for Organismal Studies (COS), University of Heidelberg, 69120 Heidelberg, Germany; gernot.poschet@cos.uni-heidelberg.de

**Keywords:** carnosinase 1, CN1, *Cndp1*, carnosine, anserine, kidney, glucose homeostasis

## Abstract

Carnosinase 1 (CN1) is encoded by the *Cndp1* gene and degrades carnosine and anserine, two natural histidine-containing dipeptides. In vitro and in vivo studies suggest carnosine- and anserine-mediated protection against long-term sequelae of reactive metabolites accumulating, e.g., in diabetes mellitus. We have characterized the metabolic impact of CN1 in 11- and 55-week-old *Cndp1*-knockout (*Cndp1*-KO) mice and litter-matched wildtypes (WT). In *Cndp1*-KO mice, renal carnosine and anserine concentrations were gender-specifically increased 2- to 9-fold, respectively in the kidney and both most abundant in the renal cortex, but remained unchanged in all other organs and in serum. Renal oxidized/reduced glutathione concentrations, renal morphology and function were unaltered. In *Cndp1*-KO mice at week 11, renal asparagine, serine and glutamine levels and at week 55, renal arginine concentration were reduced. Renal heat-shock-protein 70 (*Hspa1a/b*) mRNA declined with age in WT but not in *Cndp1*-KO mice, transcription factor heat-shock-factor 1 was higher in 55-week-old KO mice. Fasting blood glucose concentrations decreased with age in WT mice, but were unchanged in *Cndp1-KO* mice. Blood glucose response to intraperitoneal insulin was gender- but not genotype-dependent, the response to intraperitoneal glucose injection was similar in all groups. A global *Cndp1*-KO selectively, age- and gender-specifically, increases renal carnosine and anserine concentrations, alters renal amino acid- and HSP70 profile and modifies systemic glucose homeostasis. Increase of the natural occurring carnosine and anserine levels in the kidney by modulation of CN1 represents a promising therapeutic approach to mitigate or prevent chronic kidney diseases such as diabetic nephropathy.

## 1. Introduction

Carnosinase 1 (CN1; EC 3.4.13.20) is a member of the M20 metalloprotease family with a catalytic domain featuring a dinuclear Zn^2+^ [1,2]. In aqueous solution CN1 is present as homodimer with narrow substrate specificity for the naturally occurring dipeptides carnosine (β-alanyl-*L*-histidine), and its derivatives anserine (β-alanyl-*N*-methylhistidine) and homocarnosine (γ-amino-butyryl-*L*-histidine) [2,3]. Carnosinase 1 (CN1) activity, defined by genetic variants, has been linked to the incidence of nephropathy in female patients with diabetes mellitus [4,5]. Carnosine scavenges carbonyls [6,7], inhibits glycation [8] and acts as an ACE inhibitor [9,10,11]. Carnosine and anserine can prevent methylglyoxal (MG)-induced advanced glycation endproducts (AGE) *N*-(1-Carboxyethyl)-*L*-lysine (CEL) formation, even though cellular uptake of carnosine in vitro is low. High quenching capacity of carnosine has been shown for acrolein and 4-hydroxynonenal (HNE) [12,13,14]. Moreover, carnosine may act as an antioxidant, albeit this action could not unequivocally be reproduced [15,16,17]. Anserine upregulates HSP70 expression under conditions of oxidative and glycating stress in tubular cells and in type 2 diabetic mice, anserine also reduces blood glucose levels and vascular permeability and proteinuria [18]. In mammals and humans carnosine and anserine are the most abundant dipeptides, with highest concentrations in muscles [1]. Plasma concentrations of carnosine and anserine are higher in rodents compared to humans, because rodents have no carnosine-degrading serum enzyme. Carnosine actions in humans may therefore predominately be exerted via local, tissue specific carnosine metabolism. In kidneys of mice and men, anserine and carnosine are present at high concentrations, with anserine concentrations exceeding those of carnosine [19]. Supplementation of carnosine in diabetic rodents consistently improved renal histology and function [20], vascular permeability [21], and wound healing [22], but also glucose homeostasis [23,24,25,26]. Whether the beneficial effects of carnosine resulted from interference with reactive metabolite-induced molecular pathomechanisms or primarily via improving glucose homeostasis is yet unclear. Initial intervention studies in (pre-)diabetic patients also yielded promising results, even though the half-life of carnosine in the human circulation is short even in subjects with low CN1 activity [27].

Taken together, interference with the metabolism of carnosine and anserine may be a promising therapeutic approach in patients with diabetes. To date, however, the precise molecular mechanisms of carnosine generation and related metabolic actions are not yet sufficiently understood. Therefore, we characterized a *Cndp1*-knockout (*Cndp1*-KO) mouse model with respect to systemic and local tissue carnosine and anserine metabolism and glucose homeostasis. 

## 2. Material and Methods

### 2.1. Mouse Strains

Animals of the strain B6;129S5-*Cndp1*^tm1Lex^/Mmucd (#032215; MMRRC UC Davis, Davis, UT, USA), harboring a targeted mutation in exons 8 and 9 of the *Cndp1* gene, generated by homologous recombination and verified by Southernblot [28], were used. These mice were cross-bred with C57BL/6J animals (Charles River, Sulzfeld, Gemany) for 10 generations and offspring homozygous for the *Cndp1* knockout, verified by PCR (https://mmrrc.ucdavis.edu/protocols/032215Geno_Protocol.pdf; supplement figure), were used in the experiments. C57BL/6J mice were used as wild type controls. Breeding scheme consisted of crossing of homozygous males with homozygous females. Generations F1 to F3 were used for the experiments. The global Cndp1-KO has been confirmed by gene expression (https://mmrrc.ucdavis.edu/lexiconphenotype.php?id=PRT333N1). Blood chemistry (including non-fasting blood glucose, cholesterol and triglyceride concentrations) have been reported by MMRRC and were not different for *Cndp1-KO* compared to WT mice (https://mmrrc.ucdavis.edu/lexiconphenotype.php?id=PRT333N1). Animals were housed at Interfaculty Biomedical Facility (IBF) at Heidelberg University with a daily cycle of 12 h light and 12 h darkness at 23 °C. Food and water were supplied ad libitum. Experiments were performed with 11- and 55-week-old *Cndp1*-KO and wild type mice. The studies were approved by the respective authorities (Regierungspräsidium Karlsruhe, Germany, 35-9165 81/G-209/16).

### 2.2. Carnosinase Activity

50 mg frozen tissue (either brain, liver, muscle, heart, lungs or kidney) was homogenized in 500 uL buffer (20 mmol/L HEPES (pH 7.2), 210 mmol/L mannitol, 70 mmol/L sucrose). Samples were centrifuged for 10 min at 10,000 g and 37.5 uL of carnosine substrate solution (1 mg/mL) was added to 150 uL sample supernatant. After incubation for 0, 10, 20 and 40 min at 30 °C with 40 uL sample per time point, the reaction was terminated and liberated L-histidine was derivatized by adding 110 uL o-phthaldialdehyde, trichloroacetic acid and Tris buffer (50 mmol/L; pH 7.5) at a 1:1:1 ratio. Fluorescence was then measured using a microplate reader (infinite M200, Tecan, Männedorf, Switzerland) with λEx 360 nm and λEm 460 nm. To inhibit residual renal carnosine degrading activity in *Cndp1*-KO mice this assay was repeated with an additionally 0.1 mmol/L bestatin added to the carnosine substrate solution [21].

### 2.3. Carnosine and Anserine Concentrations

Carnosine concentrations were measured fluorometrically using HPLC. 50 mg of frozen tissue (either brain, liver, muscle, heart, lungs or kidney) was homogenized in 500 uL buffer (20 mmol/L HEPES; pH 7.2, 210 mmol/L mannitol, 70 mmol/L sucrose) and centrifuged for 10 min at 10,000× *g*. 200 uL supernatant was incubated with 50 uL of sulfosalicylic acid (or 100 uL serum with 25 uL sulfosalicylic acid) for 30 min on ice and centrifuged for 5 min at 10,000× *g*. Supernatants were then diluted 1:3 with borate buffer (400 mmol/L; pH 9.5) and 180 uL of each sample was derivatized with 180 uL carbazole-9-carbonyl chloride (0.567 mg/mL in acetone) for 90 s and 108 uL of 150 mmol/L hydroxylammonium chloride with 68 mmol/L NaOH and 2% methylthioethanol (*v*/*v*) for 3 min, before the reaction was terminated with 252 uL of acetonitrile in acetic acid in 8:2 ratio (*v*/*v*). Samples were applied on a HPLC column (Jupiter column C18, 300 Å, 5 µm particle size, 250 × 4.6 mm, Phenomenex, Aschaffenburg, Germany) and carnosine and anserine were measured by fluorescence detection (RF-20A, Shimadzu, Kyoto, Japan) at λex 287 nm/λem 340 nm. The mobile phase consisted of 82% solution A (50 mmol/L acetate buffer in distilled water; pH 4.37) and 18% solution B (a mixture of acetonitrile, methanol and tetrahydrofuran 70:25:5 (*v*/*v*/*v*)). Solutions were degassed prior to use. All samples were measured at least twice, and one sample was spiked with standards to definitively identify each analyte [19].

### 2.4. Blood Glucose

Blood glucose levels of mice were determined after a fasting period of 5 h with a glucometer (Roche, Basel, Switzerland). Experiments and daily treatment were performed during “rest” period at comparable timeslots during the day. Blood was drawn by puncture of the tail vein.

### 2.5. Hematological Analysis

Hemoglobin (Hb) concentration, red blood cell (RBC) count and erythrocyte indexes (MCV, MCH) were measured on a IDEXX Procyte dx automated blood cell analyzer (Idexx Laboratories).

### 2.6. Food Intake

Food intake of mice was determined by measuring decrease in food cup weight over a time-span of 3 to 4 days. The difference was divided by the number of mice per cage (1–2 mice per cage).

### 2.7. Localization of Carnosine, Anserine, GSH and GSSG

Localization of carnosine, anserine, glutathione (GSH), and glutathione disulfide (GSSG) was conducted in kidney tissue sections of 55-week-old *Cndp1*-KO and wildtype mice by Mass Spectrometry Imaging (MSI). Serial sections of the investigated kidney tissue were collected on a cryomicrotome at 14 µm thickness and thaw-mounted onto indium tin oxide-coated (ITO) glass slides. The ITO slides were scanned using a flatbed scanner and dried in a desiccator until analysis. Kidney tissue sections for measurement in positive ion mode were then sprayed with DHB matrix (20 mg/mL, 60% methanol, 0.1% trifluoroacetic acid) using a TM-Sprayer (HTX Technologies, Chapel Hill, NC, USA). For measurements in negative ion mode 9-aminoacridine (9-AA) matrix (10 mg/mL in 70% ethanol) was used instead. MSI analysis was carried out using a solariX XR 7T FT-ICR mass spectrometer (Bruker Daltonik, Billerica, MA, USA). Detection of anserine and carnosine was conducted in positive ion mode using Continuous Accumulation of Selected Ions (CASI) with multiple *m*/*z*-windows with a width of ±5 from the theoretical *m*/*z* of the target compound. MALDI-MSI analysis of GSH and GSSG was performed in negative ion mode using full scan mode of the *m*/*z* range 100–3000. *m*/*z*-images were generated using flexImaging 5.0 (Bruker Daltonik, Billerica, MA, USA). To alleviate variations in pixel-to-pixel intensity, the spectrum of each pixel was normalized against the root mean square intensity of all its data points. The investigated compounds (anserine, carnosine, GSH, GSSG) were annotated on the basis of accurate mass (<0.002 Da in CASI mode and <0.005 Da in full scan mode) and by matching to a reference standard spotted on the same ITO slide off-tissue [29].

### 2.8. Histological Staining

Representative formalin-fixed paraffin-embedded (FFPE) kidney and muscle specimens were cut in 4–6 µm sections using a HM 340E Rotary Microtome (Thermo Scientific Fisher, Waltham, MA, USA) and stained with hematoxylin and eosin (HE) and acid fuchsin orange G (AFOG) according to standard protocols. Additionally, a periodic-acid Schiff reaction (PAS) was performed. Images were acquired at 20-fold magnification using a Hamamatsu NanoZoomer Digital Pathology system (Hamamatsu Photonics, Hamamatsu, Japan).

### 2.9. Glomerular Filtration Rate

Glomerular filtration rate (GFR) was determined by transcutaneous fluorescence measurement of the clearance of fluorescein isothiocyanate (FITC)-sinistrin. One day before measurement, mice were anesthetized (initial: 5% isoflurane and 600 mL/min O_2_ for 1 min; following: 2.5% isoflurane and 200 mL/min O_2_) and the side of the back was shaved and further treated with depilation cream. The next day, mice were again anesthetized and the measuring device with a light emitting diode (λex 480 nm) for excitation of FITC and a photodiode (λem 520 nm) for fluorescence detection was attached to the shaved area. After 2 min for detecting the measurement’s baseline, 60 uL FITC-sinistrin solution (50 mg/mL in sterile saline solution; Fresenius-Kabi, Bad Homburg, Germany) was injected retro-orbitally and mice were placed back in their cages. After 2 h, measuring devices were removed and data were analyzed with MPD-Lab (Mannheim Pharma and Diagnostics, Mannheim, Germany). GFR was calculated out of the half-lifes of the FITC-sinistrin excretion in a semi-empirical approach with a one-compartment model.

### 2.10. Albumin Creatinine Ratios

Albumin and creatinine were determined from spot urine collections of the mice. Albumin was measured by Mouse Albumin ELISA (Alpco, Salem, MA, USA) and creatinine with Creatinine Colorimetric/Fluorometric Assay Kit (BioVision, Milpitas, CA, USA) according to the manufacturer’s manual.

### 2.11. Intraperitoneal Insulin Tolerance Test (IPITT) and Intraperitoneal Glucose Tolerance Test (IPGTT)

Prior to both tests, mice were fasted for 5 h and further deprived of food and water for the duration of the experiments. The animals were restrained, and tail veins were punctured. Blood glucose was measured with a glucose meter (Aviva Accu Check, Roche, Mannheim, Germany), representing baseline levels. For the insulin tolerance test, the insulin stock solution (40 U/mL; Insuman Rapid, Sanofi-Aventis, Frankfurt, Germany) was diluted to 1 U/mL in sterile saline solution. Mice were injected intraperitoneal with 1 U/kg body weight. For the glucose tolerance test, mice were injected with 2 g/kg body mass of D-glucose (20% w/v in sterile saline solution). For both experiments blood glucose was measured after 5, 10, 15, 20, 25, 30, 45, 60, 80, 100 and 120 min after injection.

### 2.12. Thiol- and Amino Acid Concentrations

Free amino acids and thiol compounds were extracted from 30 mg kidney tissue with 0.3 mL of 0.1 mol/L HCl in an ultrasonic ice-bath for 10 min. The resulting homogenates were centrifuged twice for 10 min at 4 °C and 16,400× *g* to remove cell debris. Total glutathione was quantified by reducing disulfides with DTT followed by thiol derivatization with the fluorescent dye monobromobimane (Thiolyte, Calbiochem, Darmstadt, Germany). For quantification of GSSG, free thiols were first blocked by NEM followed by DTT reduction and monobromobimane derivatization. GSH equivalents were calculated by subtracting GSSG from total glutathione levels. Derivatization was performed as described in Wirtz et al. [30]. For ultra performance liquid chromatography-fluorescence detector (UPLC-FLR) analysis an Acquity H-class UPLC system coupled to an Acquity FLR detector (Waters, Milford, CT, USA) was used. Separation was carried out using a binary gradient of buffer A (100 mmol/L potassium acetate, pH 5.3) and solvent B (acetonitrile) with the following gradient: 0 min 2.3% buffer B; 0.99 min 2.3%, 1 min 70%, 1.45 min 70%, and re-equilibration to 2.3% B in 1.05 min at a flow rate of 0.85 mL min^−1^. The column (Acquity BEH Shield RP18 column, 50 mm x 2.1 mm, 1.7 µm, Waters, Milford, USA) was maintained at 45 °C and sample temperature was kept constant at 14 °C. Monobromobimane conjugates were detected by fluorescence at 480 nm after excitation at 380 nm. Data acquisition and processing was performed with the Empower3 software suite (Waters, Milford, CT, USA).

Determination of proteinogenic amino acid levels was done as described in Weger et al. [31]. Beta-Alanine content was analyzed after specific labeling with the fluorescence dye AccQ-Tag™ (Waters, Milford, CT, USA) according to the manufacturer’s protocol using an Acquity I-class UPLC system coupled to a VION Ion Mobility Separation QTof (Waters, Milford, CT, USA). Separation was carried out using a Cortecs C18 column (100 mm × 2.1 mm, 1.6 µm, Waters) at 40 °C. The mobile UPLC phase consisted of binary gradients of ACN with 0.1% formic acid (B) and 0.1% aqueous formic acid (A), flowing at 0.5 mL min^−1^. Analytes were initially eluted with 98% A and A was decreased linearly to 35% over 6 min. After this, the column was washed with 85% B for 2 min and re-equilibrated under the initial conditions for 2 min. Measurements were performed with an ESI source operated in positive mode (1.00 kV capillary voltage; source temperature 120 °C, desolvation temperature 500 °C; observed *m*/*z* 260.103 Da; observed CCS value 161.84). Unifi software (version 1.9.3, Waters, Milford, CT, USA) was used to control the instrument and to acquire and process the MS data.

### 2.13. Expression Analysis by qPCR

Kidney tissue was homogenized using the Ball mill MM400 from Retsch with the corresponding tissue lyser adaptor sets from QIAGEN. RNeasy Mini kit from QIAGEN was used for RNA purification. Isolated mRNA was transcribed to cDNA (High Capacity cDNA Reverse Transcription Kit, Thermo Fisher Scientific, Waltham, MA, USA). Samples were analyzed using SYBR Green (PowerUP SYBR Green Master mix, Thermo Fisher Scientific) based Quantitative RT-PCR (StepOne Plus RT-PCR System, Applied Biosystems, Foster City, CA, USA). Hypoxanthine guanine phosphoribosyl transferase (Hprt) served as house-keeping gene (Table 1). Data were analyzed using the ddCT Method.

### 2.14. Protein Oxidation

To measure introduction of carbonyl groups into proteins, OxyBlot Protein Oxidation Detection Kit (Merck, Darmstadt, Germany) was used. Kidney tissue samples of *Cndp1*-KO and WT mice were analyzed according to the manufacturer’s manual [31].

### 2.15. Statistical Analysis

Data were obtained from at least three independent experiments and are given as mean with standard deviation (SD). Error of relative differences of means is given as standard error (SE). Relative expression changes are given with a range of fold change. Statistical analysis was performed with R, a language for computational statistics and graphics (https://www.r-project.org). Means were compared by student’s t-test. Generalized additive mixed models (GAMM) from R package “mgcv” were used to model relative blood glucose levels over time with respect to sex (male/female) and genotype (*Cndp1*-KO and wildtype). Analysis of variance (ANOVA) was used to analyze effects of sex (female, male), genotype (*Cndp1*-KO and wildtype) and age (11 weeks and 55 weeks of life) on blood glucose levels after fasting. ANOVA post hoc comparisons were computed using estimated marginal means with R package “emmenas”. *p*-values of <0.05 were considered significant.

## 3. Results

### 3.1. Renal Carnosine and Anserine Metabolism

Maximal *ex-vivo* carnosine degrading activity of renal tissue CN1 (V_max_ after addition of 1 mM carnosine) was higher in 55-week-old compared to 11-week-old WT mice (3.3 ± 2.7 vs. 1.4 ± 0.8 nmol/(mg*h)), without any gender-specific differences. In *Cndp1*-KO mice, renal carnosine-degrading activity was severely reduced (0.7 ± 1.3 and 0.2 ± 0.2 nmol/(mg*h) at week 55 and 11; *p* = 0.001 versus WT for both age groups) and could be abolished by addition of bestatin, an inhibitor of carnosinase 2 (CN2). Carnosine degrading efficiency in WT mice was not affected by addition of bestatin.

In 11-week-old *Cndp1*-KO mice renal carnosine and anserine concentrations were 4- and 2- fold higher compared to their age-matched WT controls and in 55-week-old KO mice 2- and 9-fold higher, respectively (Table 2, Figure 1). Gender-specific differences in histidine dipeptide concentrations were observed for carnosine in 11-week-old WT and 55-week-old KO mice (Table 2). These genotype and gender specific differences in renal carnosine and anserine abundance were confirmed by MALDI-MSI in 55-week-old mice (Figure 2). Both dipeptides were mainly localized in the renal cortex. An age dependent decline in renal carnosine and anserine concentrations was observed in female WT mice and in male *Cndp1*-KO mice.

### 3.2. Non-Renal Tissue Carnosine and Anserine Metabolism

Carnosine-degrading activity was consistently measured in renal tissue of WT mice, but not in muscle, brain, liver, pancreas, lungs, heart or serum; a tissue carnosine synthase activity assay has not yet been established. Carnosine and anserine were detected in brain, liver, muscle, heart, lungs and serum (Table 3) of WT and KO mice. In both genotypes, the highest carnosine and anserine values were found in muscle, with 5–10 times higher concentrations compared to the other organs. The carnosine-to-anserine ratio was highest in brain (3-to7-fold higher carnosine compared to anserine concentrations) and lowest ratios were found in the heart (equal concentrations in 11-weeksold and 2 to 3-fold higher anserine concentrations in 55-week-old mice). Serum carnosine and anserine concentrations measured at week 55 were substantially lower than respective muscle concentrations but largely in the same range as in other organs. In pancreas carnosine and anserine could not consistently be measured by HPLC.

In *Cndp1*-KO mice, no increase in organ carnosine or anserine concentrations compared to WT mice were detected. At week 11, lower carnosine concentrations were measured in liver and muscle and lower anserine concentrations in the heart of *Cndp1*-KO mice; at week 55 all tissue (brain, liver, muscle, heart, lungs or kidney) dipeptide concentrations were comparable between WT and KO mice. Brain carnosine concentrations increased with age in both genotypes (Table 3).

### 3.3. Amino Acid Profile

Renal amino acids were measured at weeks 11 and 55. In *Cndp1*-KO mice, renal asparagine, glutamine and serine concentrations were reduced by 30, 45 and 30%, respectively (*p* = 0.042/0.001/0.05; Figure 3A) at week 11. At 55 weeks, renal arginine concentrations were reduced by 20% in *Cndp1*-KO compared to WT mice (*p* ≤ 0.05; Figure 3B). ß-alanine, the rate limiting substrate of carnosine synthesis, measured at week 11, was not different between genotypes in kidneys, muscle and serum (data not shown).

### 3.4. Kidney Morphology and Function

Staining with hematoxylin and eosin (HE), with acid fuchsin orange G (AFOG) and periodic-acid Schiff reaction (PAS) of renal tissue sections revealed no structural alterations in *Cndp1*-KO and WT mice (Figure 4A–D), i.e., no basal membrane thickening, podocyte loss or mesangial hypertrophy. Deposition of PAS positive material in glomeruli measured by densitometry did not differ between *Cndp1*-KO and WT mice (Figure 4E). Glomerular filtration rate, assessed by transcutaneous measurement of FITC-sinistrin clearance over time, was not different between 55-week-old *Cndp1*-KO and WT mice (Figure 4F). Albumin to creatinine ratio was in the same range in *Cndp1*-KO and WT mice (data not shown).

### 3.5. Oxidative Stress and Inflammation Marker

Renal concentrations of oxidized (GSSG) and reduced glutathione (GSH) and its precursor γ-glutamylcysteine (γ-EC) were within the same range for *Cndp1*-KO and WT mice at week 55 (Figure 5A). GSH and GSSG were mainly localized in the medulla of the kidney as indicated by MALDI-MSI analysis (Figure 5B). Renal protein carbonyl levels determined by OxyBlot were low and not different between WT and KO animals (data not shown). Renal Hsp70 (Hspa1a and Hspa1b) mRNA expression decreased with age in WT mice (*p* = 0.04 and *p* = 0.01) but did not decrease with age in *Cndp1*-KO mice (Figure 6). Renal mRNA expression of heat shock factor 1 (Hsf1), the major transcription factor for heat shock proteins, was similar at weeks 11 and 55 in WT mice but increased in 55-week-old *Cndp1*-KO mice (*p* = 0.01; Figure 6). ADP/ATP ratios were not different in 55-week-old *Cndp1*-KO and WT mice (5.3 ± 0.4 vs. 5.7 ± 0.4; *p* = n.s.), suggesting comparable energy metabolism. A hemogram, including differential blood count, hematocrit, mean corpuscular volume and hemoglobin, showed no difference between WT and *Cndp1*-KO mice.

### 3.6. Glucose Homeostasis

At 11 weeks, blood glucose concentrations were higher in male WT mice than in respective females (169 ± 19 vs. 146 ± 23 mg/dl; *p* = 0.003) and decreased with age in both genders (Figure 7; *p* = 0.0001 in males and *p* = 0.016 in females). In contrast, in *Cndp1*-KO mice, fasting blood glucose were comparable in both genders and did not change with age (*p* = ns). 55-week-old female *Cndp1*-KO mice had higher fasting blood glucose levels than respective WT controls (*p* = 0.006). In 11-week-old male *Cndp1*-KO mice fasting blood glucose concentrations were lower than in respective male WT mice (*p* = 0.002).

The intraperitoneal insulin tolerance test (IP-ITT) overall did not result in differences in the blood glucose profile over 120 min in 11-week-old *Cndp1*-KO versus WT mice. In female *Cndp1*-KO mice the blood glucose decline was less pronounced than in female WT mice (Figure 8). GAMM analysis taking genotype and gender into account, identified gender as a predictor of blood glucose response during the IP-ITT (*p* = 0.027) but not genotype. The IP-ITT at week 55 demonstrated a stronger decline in glucose concentrations after 10, 15, 20 and 60 min in *Cndp1*-KO mice. The difference was evident in males only (Figure 8). In line with this, GAMM analysis taking genotype and gender into account, identified gender as strong predictor of ITT blood glucose levels (*p* = 0.0009), but not genotype (*p* = 0.098). In GAMM analysis limited to genotype, genotype was a weak predictor of ITT blood glucose levels (*p* = 0.045).

Intraperitoneal glucose tolerance tests (IP-GTT) performed at age of 11 and 55 weeks resulted in similar increased blood glucose concentrations relative to baseline blood glucose in both gender and genotypes over 120 min (Figure 9). GAMM analysis did neither reveal an effect of genotype nor of gender on blood glucose levels over 120 min of IP-GTT at weeks 11 and 55.

Plasma insulin concentrations determined at week 55 were higher in WT male vs. female mice (male WT: 1.61 ± 0.2 vs. female WT: 0.92 ± 0.29, *p* = 0.025; male Cndp1-KO: 1.93 ± 0.92 vs. female Cndp1-KO 1.36 ± 1.15; *p* = 0.6) and not different between WT and Cndp1-KO mice (WT: 1.27 ± 0.43; Cndp1-KO: 1.59 ± 0.99; *p* = 0.48).

### 3.7. Body and Organ Weight in Cndp1-KO and WT Mice

Body weight was similar in male and female *Cndp1*-KO and WT mice at the age of 11 weeks (Table 4). Until the age of 55 weeks, body weight increased by 44% in *Cndp1*-KO mice, but only by 27% in WT mice (*p* = 3.8*10^−10^ and *p* = 0.003). The genotype dependent difference in body weight were similar in both genders and 13 and 16% higher in male and female *Cndp1*-KO mice compared to respective WT mice (*p* = 0.014 and *p* = 0.005 versus WT, *p* = ns between gender). The food-intake (g/mouse/24 h) was significantly higher (*p* = 0.008) in *Cndp1*-KO compared to WT mice (Table 5). Except for the lungs, all organ weights were higher in *Cndp1*-KO compared to WT at week 55. Relative to body weight brain and lung weight were reduced, and spleen weight increased in *Cndp1*-KO animals. Hematoxylin and eosin (HE) staining demonstrated unaltered muscle morphology (data not shown).

## 4. Discussion

Carnosine scavenges carbonyls [6,7], inhibits glycation [8], prevents AGE formation and has antioxidant actions. Supplementation of carnosine in diabetic rodents increases renal carnosine concentrations [32] and consistently preserves renal morphology and function [20]. In humans carnosinemia, due to CN1 deficiency may be a non-disease; whereas accumulation of carnosine should protect against long-term sequelae of reactive metabolites accumulating, e.g., in diabetes and chronic renal failure [3,20] and may prevent cancer development [33]. Efficacy of therapeutic carnosine supplementation in humans, however, is limited by a highly active serum CN1 largely preventing major increases in serum carnosine concentrations [27]. The human kidney expresses carnosine-synthetase and CN1 in glomeruli and tubular cells, CN1 activity is increased in diabetic kidneys [19,21]. Modulation of kidney carnosine metabolism to increase tissue carnosine and anserine concentrations is a promising therapeutic concept. Before aiming for such therapeutic interventions, the local and systemic metabolic consequences require detailed analyses. We therefore studied a mouse model with high CN1 activity in the WT kidney but none in the muscle, brain, liver, pancreas, lungs, heart or serum. The respective global *Cndp1*-KO resulted in a gender-specific, selective 2- to 9-fold increase in renal carnosine and anserine concentrations, while respective concentrations were unchanged or slightly decreased in all other organs and blood. In these *Cndp1*-KO mice, renal carnosine-degrading efficiency was severely reduced, indicating that CN2 hardly compensates for CN1 activity as previously discussed [1]. Further, we confirmed previous findings [2] that CN1 activity in this mouse model is only detectable in the kidney. Thus, this mouse model is uniquely suited to study the impact of renal histidine-dipeptides in the development and progression of renal disease.

The persistently high renal concentrations of carnosine and anserine did not result in any renal morphological and/or functional abnormalities in comparison with WT mice. Putative quenching capacity against reactive metabolites, such as acrolein and HNE [34] could not be demonstrated, the degree of carbonylation was low in all mice. Renal GSSG-to-GSH ratio and the precursor γ-EC were unaltered. Recent studies have shown that heat shock response contributes to establishing a cytoprotective state in a wide variety of human diseases, such as diabetes or cancer. At least part of the protective effect of carnosine might be related to induced expression of members of the heat shock family, such as hsp70, Hsp72 and/or hsp90 [35,36,37]. Age-associated decline of renal HSP70 expression was prevented in 55-week-old *Cndp1*-KO mice, Hsf1 increased. HSP70 protects against oxidative stress, participates in disposal of damaged or defective proteins, improves protein integrity and inhibits apoptosis [38]. Hsf1 is the major transcription factor of HSPs. In human proximal tubular cells HSP70 expression is induced by oxidative and glucose stress in the presence of anserine, but not carnosine [18]. Whether the *Cndp1*-KO related changes in cellular protective mechanisms and the renal accumulation of carnosine and anserine provide a significant protection against diseases such as diabetic nephropathy, remains to be investigated in respective animal models.

In addition to the local renal metabolic changes, the *Cndp1*-KO, which was functionally confined to the kidneys, resulted in modifications of systemic glucose homeostasis. While there was an age- dependent decline in fasting blood glucose concentrations in the C57BL/6J WT mice as previously described [39], blood glucose remained unchanged in the respective KO mice. In line with this, the genotype predicted the response to intraperitoneal insulin injection in 55-week-old mice. However, when gender was taken into account, the observed differences were predicted by gender but not genotype. The response to intraperitoneal glucose was similar in all age and gender groups. Thus, the overall effect of *Cndp1*-KO on glucose homeostasis was not pronounced; the molecular mechanisms are uncertain. Carnosine treatment of diabetic mice resulted in improved blood glucose homeostasis. In transgenic diabetic mice overexpressing human CN1 (db/db) [23], in BTBR ob/ob mice [24], in nephrectomized rats and in BALB/cA mice with streptozotocin (STZ)-induced diabetes type-1 [40,41] blood glucose and HbA1c concentrations were reduced by carnosine supply. Studies in STZ-treated mice indicated that carnosine administration modulates the insulin gene products in the pancreas, preserving insulin gene expression from STZ challenge [42]. *Cndp1* mRNA and CN1 activity could not be detected in the pancreas of our mice. Even though an impact of the proteolytic activity of the pancreatic tissue, cannot be excluded, our findings suggest that the alterations of glucose homeostasis by carnosine may be exerted via different mechanisms in the pancreas or indirectly, e.g., via metabolic effects in the kidney. Renal gluconeogenesis contributes 20–25% to whole-body glucose production [36]. Renal glutamine concentration, an important substrate for gluconeogenesis in the kidney, were halved in our 11-week-old *Cndp1*-KO mice. Thus, the renal *Cndp1*-KO may have resulted in altered renal gluconeogenesis rate. As we found HSP70 mRNA levels to be preserved in kidneys of 55-week-old *Cndp1*-KO mice, a preserved protein quality control system could also contribute to sustained endocrine pancreas function and insulin secretion.

The impact of reduced renal asparagine and serine concentrations in young and of arginine in old *Cndp1*-KO mice is uncertain. Asparagine synthetase is present in most mammalian organs, and is required for brain function and the synthesis of ammonia. Serine is important in the biosynthesis of purines and pyrimidines, and is a precursor to several amino acids and of metabolites such as sphingolipids and folate. Renal synthesis of serine only accounts for 5–7% of total body turnover of serine [43]. Arginine is a conditionally indispensable amino acid, synthesized mainly in the liver and kidney and is a precursor of nitric oxide and of urea synthesis in liver and kidney. It has immunotrophic and strong insulinogenic effects [43]. Arginine may become limiting under conditions of severe metabolic stress and the kidney may become an important arginine producer. Failure to produce sufficient quantities of arginine was reported to occur in patients with renal failure [44]. Further studies are required, in particular in mice with renal diseases, to demonstrate whether the *Cndp1*-KO associated changes in specific amino acids are of pathophysiological relevance. In the 11-week-old mice amino acid profiling was performed in 4 mice per group only, which increases the risk of a type II error, i.e., of missing differences between the two genotypes.

Higher body weight was reached in both female and male *Cndp1*-KO mice until week 55 and was associated with increased absolute weights of all organs except the lung; relative to body weight spleen weight was also increased. The higher body weight gain in *Cndp1*-KO mice is probably caused by higher food intake as compared to WT mice. Carnosine supplemented pigs achieve greater body weight gain together with an increased food intake [45] and carnosine treated STZ mice experience 50% less body weight loss than STZ mice without carnosine supplementation [42]. *Cndp1*-KO zebra fish have increased carnosine concentrations, and exhibit less body weight gain with high-fat diet than the respective WT fish [46] and a combination of genetic variations in CNDP1 and CNDP2 were associated with obesity risk in Japanese men [47]. A recent meta-analysis of 23 clinical trials suggests reduced waist circumference, HbA1c, and fasting glucose levels with histidine-dipeptide supplementation [48]. *In vitro*, carnosine scavenges glucolipotoxins, palmitic and oleic acid and increases myocyte glucose uptake under hyperglycemic conditions [48]. Thus, there is evidence from different experimental models and from clinical trials that suppression of CN1 activity with consecutive increase in carnosine tissue concentrations and carnosine supplementation, respectively, modifies food intake, glucose homeostasis and body weight gain.

In summary, a global *Cndp1*-KO in mice results in a selective and marked gender- and age-specific increase in renal carnosine and anserine concentrations, in higher renal Hsp70 expression and an altered renal amino acid profile. Blood glucose homeostasis and the response to intraperitoneal insulin is gender specifically modified, food intake and body weight gain increased. These findings suggest a significant local and systemic metabolic function of renal CN1. Suppression of renal CN1 activity and the subsequent great increase in renal carnosine and anserine concentrations represents a promising therapeutic approach to mitigate or prevent chronic kidney diseases such as diabetic nephropathy.

## Figures and Tables

**Figure 1 ijms-21-04887-f001:**
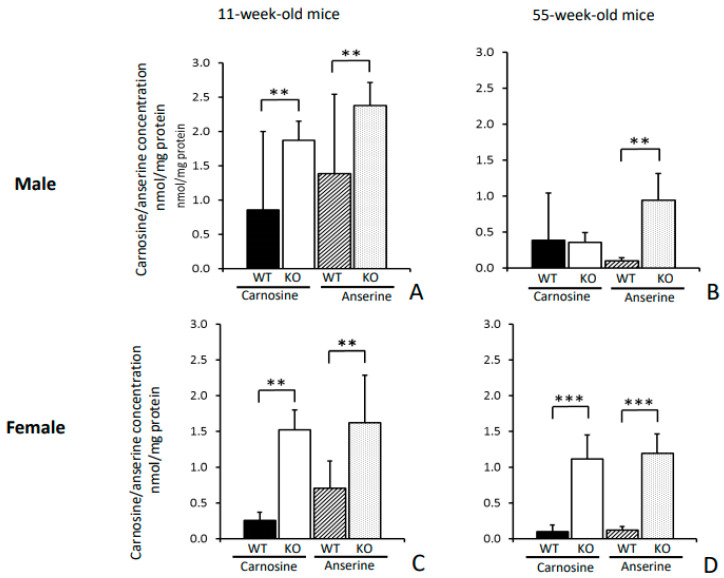
Renal carnosine and anserine concentrations. For male (**A**,**B**) and female (**C**,**D**) mice carnosine and anserine concentrations decreased with age and were higher in *Cndp1*-KO mice than in WT mice (each *n* = 4). In 55-week-old male mice (**B**), carnosine concentrations were not different between *Cndp1*-KO and WT mice. **: *p* < 0.01; ***: *p* < 0.001.

**Figure 2 ijms-21-04887-f002:**
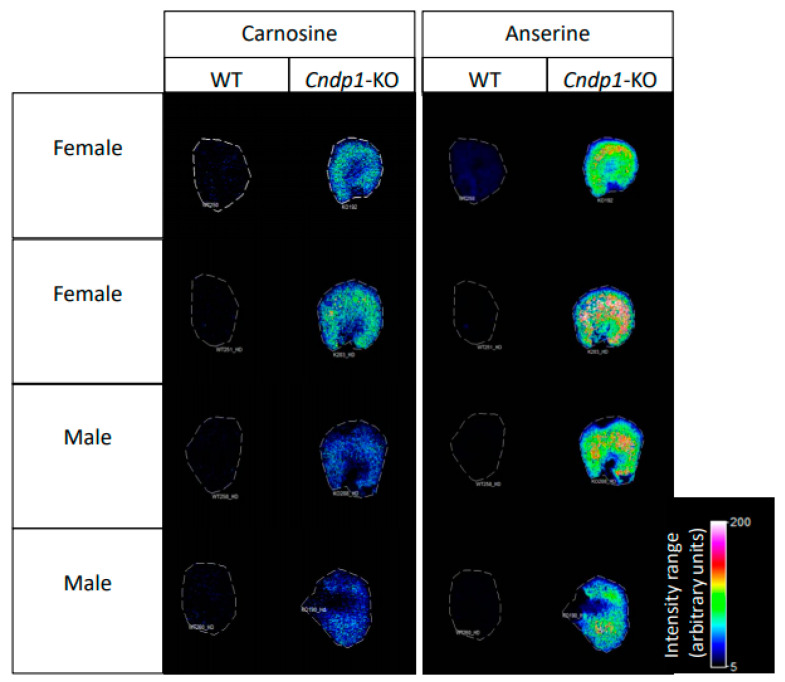
MALDI-MS Imaging of renal carnosine and anserine concentrations. MALDI-MSI confirmed increased carnosine and anserine levels in two female and 2 male, 55-week-old *Cndp1*-KO mice, as compared to respective WT mice. Both dipeptides are primarily localized in the kidney cortex.

**Figure 3 ijms-21-04887-f003:**
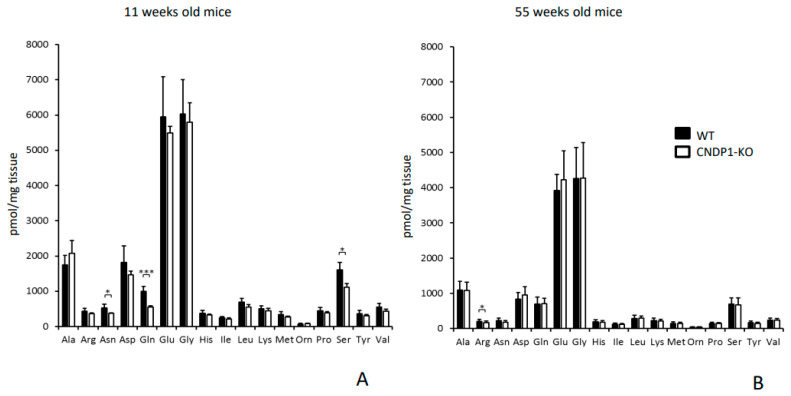
Amino acid profiles of kidneys measured by UPLC of 11- (**A**) and 55- (**B**) week-old *Cndp1*-KO (*n* = 4 and *n* = 14) and WT mice (*n* = 4 and *n* = 14). In 11-week-old mice, asparagine, glutamine and serine were decreased in *Cndp1*-KO mice compared to WT mice. In 55-week-old mice, arginine was decreased in Cndp1-KO compared to WT mice *: *p* < 0.05; ***: *p* < 0.001.

**Figure 4 ijms-21-04887-f004:**
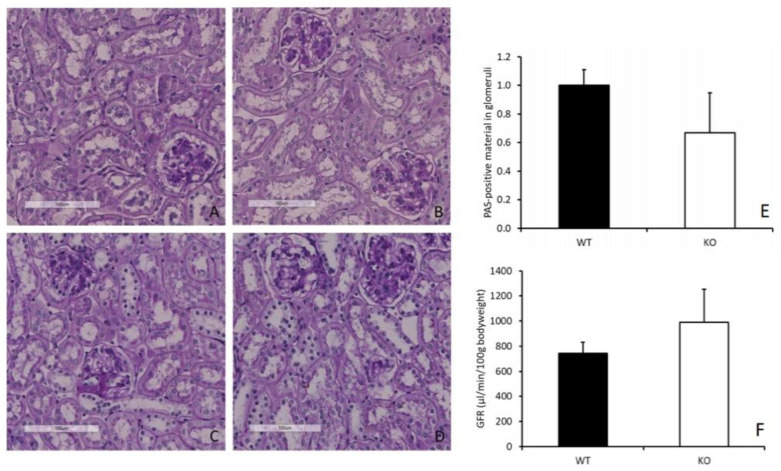
Kidney morphology and function was similar in 55-week-old *Cndp1*-KO and WT mice. Representative kidney specimens of male (**A**) and female (**B**) *Cndp1*-KO mice and male (**C**) and female (**D**) WT mice with PAS reaction showed no alterations. Deposition of PAS positive material in glomeruli of *Cndp1*-KO (*n* = 4) and WT mice (*n* = 4) was not different (**E**). Glomerular filtration rate (GFR) was also unaltered in *Cndp1*-KO mice (*n* = 11) compared to age-matched WT mice (*n* = 4) (**F**).

**Figure 5 ijms-21-04887-f005:**
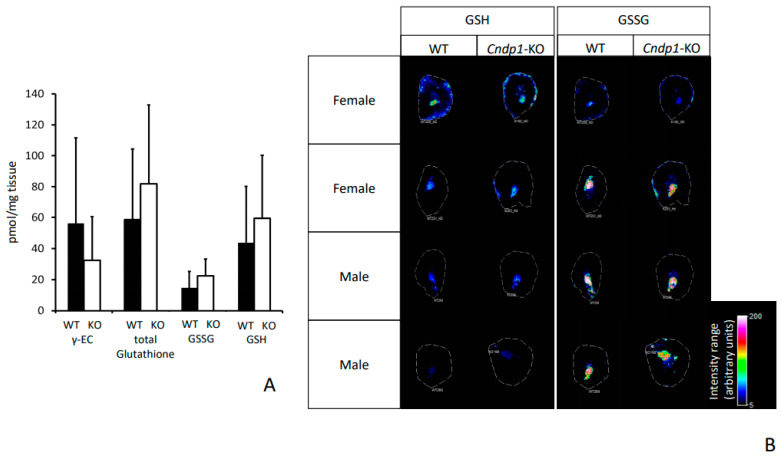
Renal glutathione (GSH) and glutathione disulfide (GSSG) concentrations and ratios were in the same range for 55-week-old *Cndp1*-KO (*n* = 14) and WT mice (*n* = 14) measured by (UPLC-FLR (**A**) and visualized by MALDI-MSI (**B**). No difference in the glutathione precursor dipeptide γ-glutamylcystein (γ-EC) was detected (**A**).

**Figure 6 ijms-21-04887-f006:**
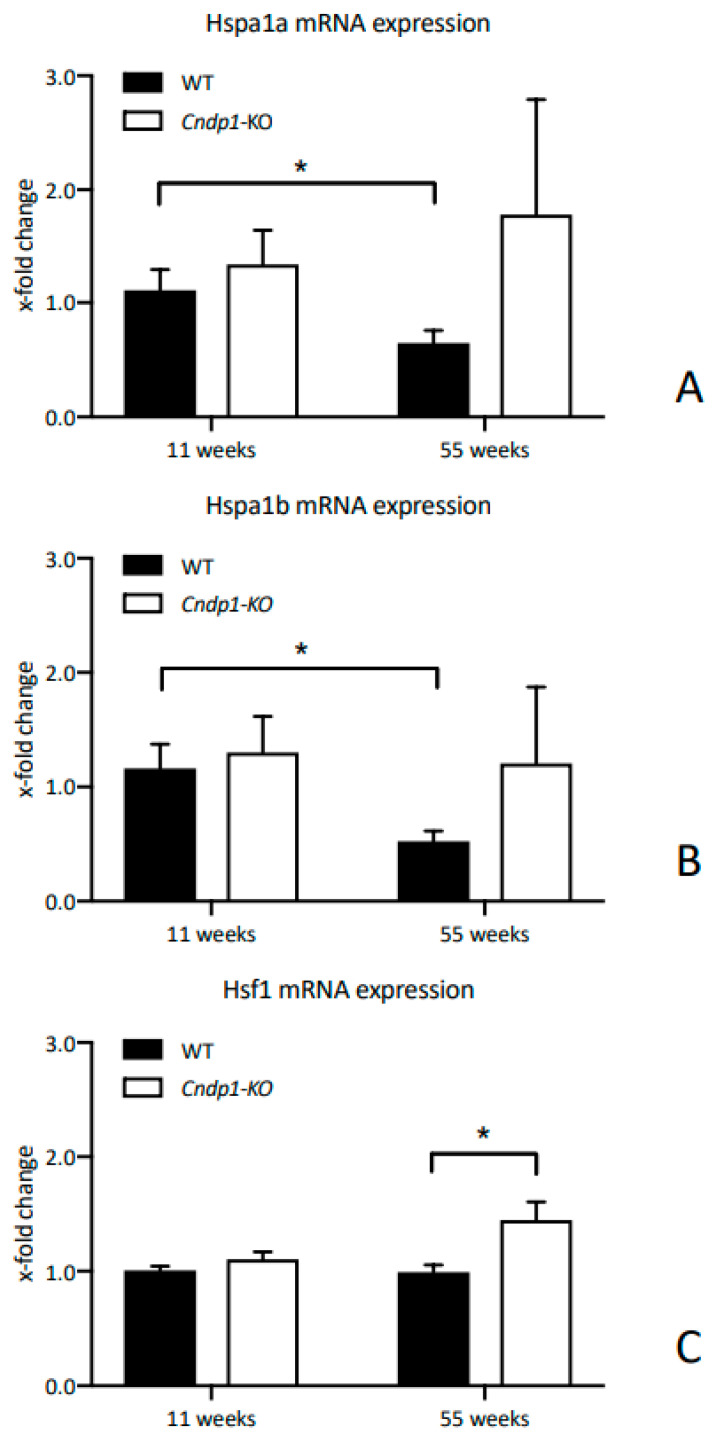
Age-associated reduction of Hspa1a (**A**) andHspa1b (**B**) mRNA expression from week 11 to week 55 in WT but not in *Cndp1*-KO mice. Hsf1 (**C**) mRNA expression was higher in *Cndp1*-KO at the age 55 weeks compared to WT mice (*n* = 10). mRNA expression was normalized to WT mice at the age of 11 weeks. Data are represented as mean ± SEM. *: *p* < 0.05.

**Figure 7 ijms-21-04887-f007:**
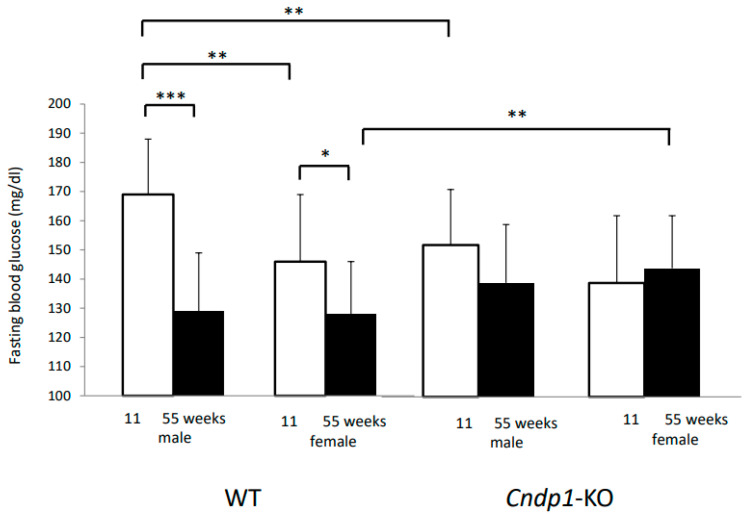
Fasting blood glucose, after 5 h fasting, decreased with age in WT mice (*p* < 0.05) but not in *Cndp1-KO* mice. (11 weeks male *Cndp1*-KO: *n* = 15; 11 weeks male WT: *n* = 17; 11 weeks female *Cndp1*-KO: *n* = 14; 11 weeks female WT: *n* = 17; 55 weeks male *Cndp1*-KO: *n* = 38; 55 weeks male WT: *n* = 12; 55 weeks old *Cndp1*-KO female: *n* = 23; 55 weeks old female WT: *n* = 17). *: *p* < 0.05; **: *p* < 0.01; ***: *p* < 0.001.

**Figure 8 ijms-21-04887-f008:**
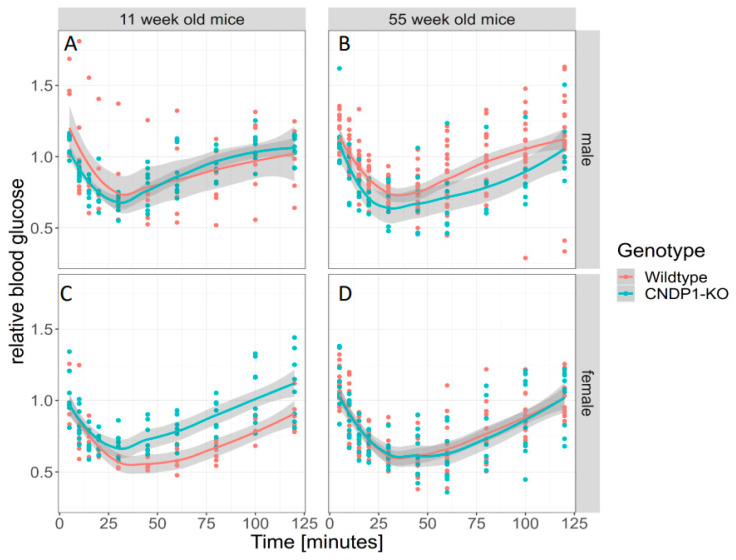
Intraperitoneal Insulin Tolerance Test (IP-ITT) in 11-week-old male (**A**) and female (**C**) *Cndp1*-KO and WT mice and in 55-week-old male (**B**) and female (**D**) *Cndp1*-KO and WT mice. Blood glucose levels are given relative to basal blood glucose level. In generalized additive mixed models (GAMM) analysis, gender (*p* = 0.027) but not the genotype predicted blood glucose levels following insulin injection, with higher values in female *Cndp1*-KO mice. (11 weeks male *Cndp1*-KO: *n* = 7; 11 weeks male WT: *n* = 8; 11 weeks female *Cndp1*-KO: *n* = 6; 11 weeks female WT: *n* = 8; 55 weeks male *Cndp1*-KO: *n* = 21; 55 weeks male WT: *n* = 7; 55 weeks old *Cndp1*-KO female: *n* = 15; 55 weeks old female WT: *n* = 10).

**Figure 9 ijms-21-04887-f009:**
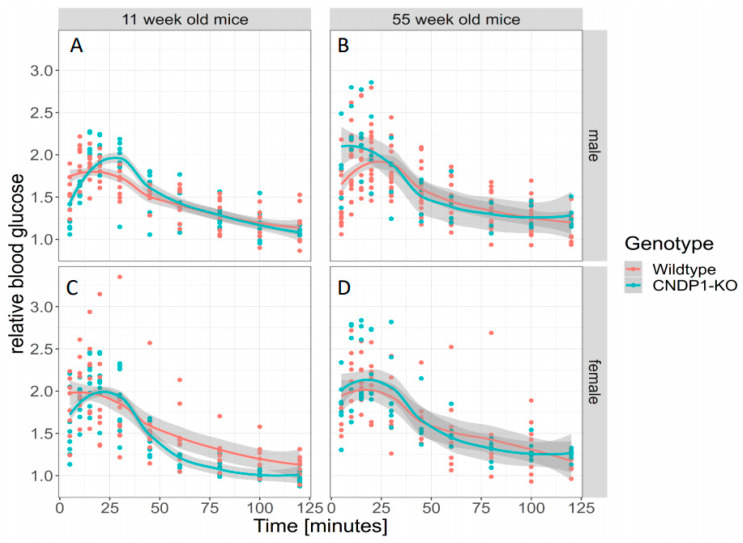
Intraperitoneal Glucose Tolerance Test (IP-GTT) in 11-week-old male (**A**) and female (**C**) *Cndp1*-KO and WT mice and in 55-week-old male (**B**) and female (**D**) *Cndp1*-KO and WT mice. Blood glucose concentrations on *y*-axis is given relative to basal blood glucose level. GAMM analysis did neither reveal an effect of genotype nor of gender on blood glucose levels over 120 min (11 weeks male *Cndp1*-KO: *n* = 8; 11 weeks male WT: *n* = 9; 11 weeks female *Cndp1*-KO: *n* = 8; 11 weeks female WT: *n* = 9; 55 weeks male *Cndp1*-KO: *n* = 17; 55 weeks male WT: *n* = 5; 55 weeks old *Cndp1*-KO female: *n* = 8; 55 weeks old female WT: *n* = 7).

**Table 1 ijms-21-04887-t001:** Primer Pairs for expression analysis by qPCR of heat shock proteins.

Gene	Forward Primer	Reverse Primer
*Hspa1a*	TGGTGCAGTCCGACATGAAG	GCTGAGAGTCGTTGAAGTAGGC
*Hspa1b*	GAGATCGACTCTCTGTTCGAGG	GCCCGTTGAAGAAGTCCTG
*Hsf1*	AACGTCCCGGCCTTCCTAA	AGATGAGCGCGTCTGTGTC
*Hprt*	TCAGTCAACGGGGGACATAAA	GGGGCTGTACTGCTTAACCAG

**Table 2 ijms-21-04887-t002:** Carnosine and anserine concentrations in kidney of 11- and 55-week-old *Cndp1*-knockout (*Cndp1*-KO) and wildtype (WT) mice (*n* = 4 for each group: WT male/female; Cndp1-KO male/female for both age groups). Significant differences (*p* ≤ 0.05) are indicated by: a = *Cndp1*-KO vs. WT; b = male vs. female; c = 55 vs. 11 weeks.

			11-Week-Old	55-Week-Old
**Wildtype**	Both gender	Carnosine [nmol/mg]	0.5 ± 0.9	0.2 ± 0.5
		Anserine [nmol/mg]	1.0 ± 0.8	0.1 ± 0.04 ^c^
	Males	Carnosine [nmol/mg]	0.9 ± 1.1 ^b^	0.4 ± 0.7
		Anserine [nmol/mg]	1.4 ± 1.1	0.1 ± 0.05
	Females	Carnosine [nmol/mg]	0.3± 0.1	0.1 ± 0.09 ^c^
		Anserine [nmol/mg]	0.7 ± 0.4	0.1 ± 0.04 ^c^
***Cndp1*-KO**	Both gender	Carnosine [nmol/mg]	1.9 ± 0.3 ^a^	0.4 ± 0.1 ^ac^
		Anserine [nmol/mg]	2.4 ± 0.3 ^a^	0.9 ± 0.4 ^ac^
	Males	Carnosine [nmol/mg]	1.7 ± 0.32 ^a^	0.7 ± 0.5 ^bc^
		Anserine [nmol/mg]	2.0 ± 0.6 ^a^	1.1 ± 0.3 ^ac^
	Females	Carnosine [nmol/mg]	1.5 ± 0.3 ^a^	1.1 ± 0.3 ^a^
		Anserine [nmol/mg]	1.6 ± 0.7 ^a^	1.2 ± 0.3 ^a^

**Table 3 ijms-21-04887-t003:** Carnosine and anserine concentrations in brain, liver, muscle, heart and lungs of 11-week-old (*n* = 8 per group) and 55-week-old *Cndp1*-KO and WT mice (*n* = 14 per group). No carnosine-degrading activity could be demonstrated in any of the organs. * = significant differences (*p* ≤ 0.05) between *Cndp1*-KO and WT, ^#^ = significant differences (*p* ≤ 0.05) between 11 and 55 weeks.

**11-Week-Old Mice**		**Brain**	**Liver**	**Muscle**	**Heart**	**Lungs**	
Wildtype	Carnosine [nmol/mg]	0.9 ± 0.3	0.2 ± 0.06	7.6 ± 1.7	0.3 ± 0.2	0.4 ± 0.4	
	Anserine [nmol/mg]	0.3 ± 0.3	0.1 ± 0.04	8.2 ± 2.4	0.3 ± 0.2	0.2 ± 0.1	
*Cndp1*-KO	Carnosine [nmol/mg]	0.7 ± 0.4	0.1 ± 0.08 ^*^	5.6 ± 0.7 ^*^	0.3 ± 0.1	0.3 ± 0.1	
	Anserine [nmol/mg]	0.1 ± 0.05	0.04 ± 0.03	7.3 ± 1.5	0.1 ± 0.1 ^*^	0.1 ± 0.04	
**55-Week-Old Mice**		**Brain**	**Liver**	**Muscle**	**Heart**	**Lungs**	**Serum**
**Wildtype**	Carnosine [nmol/mg]	1.4 ± 0.6 ^#^	0.1 ± 0.1 ^#^	8.5 ± 4.5	0.4 ± 0.6	0.7 ± 0.8	1.8 ± 0.8
	Anserine [nmol/mg]	0.2 ± 0.1	0.1 ± 0.1	11.1 ± 5.2	1.4 ± 2.7	1.7 ± 2.1 ^#^	0.7 ± 0.3
***Cndp1*-KO**	Carnosine [nmol/mg]	2.3 ± 3.3 ^#^	0.1 ± 0.1	8.4 ± 4.9	0.5 ± 0.6	0.6 ± 0.8	1.7 ± 0.8
	Anserine [nmol/mg]	0.1 ± 0.1	0.1 ± 0.1 ^#^	9.9 ± 5.2	1.0 ± 1.5	0.9 ± 1.6	1.5 ± 0.5

**Table 4 ijms-21-04887-t004:** Body weights of 11- and 55-week-old *Cndp1*-KO (*n* = 16 and *n* = 48) and WT mice (*n* = 18 and *n* = 18). Until the age of 55 weeks body weight increased by 44% in *Cndp1*-KO mice, but only by 27% in WT mice (*p* = 0.003). BW = body weight.

		11-Week-Old (g)	55-Week-Old (g)
**Wildtype**	Males and Females	22.5 ± 2.9	28.7 ± 5.5
	Males	25.2 ± 1.3	32.3 ± 5.5
	Females	19.9 ± 0.6	26.5 ± 3.5
***Cndp1*-KO**	Males and Females	23.5 ± 3.8	33.8 ± 5
	Males	26.7 ± 2.3	36.6 ± 3.9
	Females	20.2 ± 1.3	30.9 ± 4.2

**Table 5 ijms-21-04887-t005:** Food intake of 11 and 55-week-old *Cndp1*-KO and WT mice. Food intake was significantly higher in Cndp1-KO as compared to WT mice (*p* = 0.008).

Weeks of Life		Food Intake (g/Mouse/24 h)
	WT	Number of Animals	Cndp1-KO	Number of Animals
20	2.90 ± 0.05	5	3.51± 0.05	6
24	3.05 ± 0.25	5	3.45 ± 0.05	6
28	3.14 ± 0.31	5	3.01 ± 0.05	6
32	2.81 ± 0.50	5	3.32 ± 0.05	6
36	2.99 ± 0.28	5	3.79 ± 0.05	6
40	3.12 ± 0.49	5	4.05 ± 0.05	3
**Mean**	3.00		3.36

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
