# Peer review of "A Global Cndp1-Knock-Out Selectively Increases Renal Carnosine and Anserine Concentrations in an Age- and Gender-Specific Manner in Mice"

_ijms, 2020, doi:10.3390/ijms21144887_

Round 1
Reviewer 1 Report
In this study Weigand et al. have investigated the role of Carnosinase 1 (CN1) on systemic and local tissue carnosine and anserine metabolism and glucose homeostasis. To this end, they developed a Cndp1-KO mouse model. They concluded that a global Cndp1-KO in mice results in a selective and marked, gender- and age-specific increase in renal carnosine and anserine concentrations.
Please include in Materials and Methods section (or as Supplementary material) detailed information about breeding strategy, genotyping (primers), number of cohorts (males and females), and the generations (F2,F3…?) of mice used in this study for metabolic studies, animal protocol number.
Carnosinase activity, carnosine and anserine concentrations, and localization of carnosine, anserine, GSH and GSSG. Please include in Materials and Methods section a reference of this method.
Because the Cndp1-KO mouse model has not been published, and this is the first report, a full panel (PCR and Western blot) of CN1 expression/levels in different tissues (kidney, skeletal muscle, liver, pancreas, adipose tissue, brain, heart, lungs….) in WT and Cndp1-KO mice needs to be performed. Confirmation by these methods of genetic deletion of CN1 is needed.
Regarding Cndp1-KO mice compared to WT mice. A complete panel of plasma biochemistry should be added (i.e. fasting and non-fasting glucose, triglycerides, cholesterol, insulin and glucagon levels). Body weight and food intake should be assessed. Characterization of these metabolic parameters are important to understand the metabolic phenotype of KO mice.
Intraperitoneal insulin tolerance test (IPITT) and intraperitoneal glucose tolerance test (IPGTT) were not conducted according to the state-of-the-art. First, mice are not deprived of water during the course of the tests. This may cause water-deprivation stress. Second, for IPITT fasting is not required, but if so, 1-2 hours is enough, but never 5 h. Third, it is stated that mice were restrained during the 2h of tests. In IPGTT and IPITT mice are not restrained, this cause stress and compromise data. Fourth, it is not necessary blood sampling at 5, 10, 15, 20, 25 min for IPGTT, 0, 15, 30… it is enough.
Table-1. Why was observed an age-dependent decline in renal carnosine and anserine concentrations in male Cndp1-KO mice? Was not genetically ablated the enzyme in kidney?
Fig7. It is well -stablished that ageing is related to insulin resistance and impaired glucose homeostasis. How do authors explain reduced fasting blood glucose levels in WT mice at 55 weeks? What are non-fasting blood glucose levels? What are plasma insulin levels? Finally, 170 mg/dL blood glucose levels are high for a 5h fasting. Any clue?
Fig8. The IP-ITT is not plotted as state-of-the-art. The fall in blood glucose levels during the IPITT is presented as a percentage of basal glucose concentration vs. time. In addition, calculations of areas under the curve provide valuable information. Dispersion of values shown in Fig8 indicates low reproducibility among different animals.
Fig9, Same comments as above. In this case, the rise and fall in blood glucose levels during the IPGTT is presented by plotting absolute glucose values (mg/dL) in the y-axis vs. time (min) in the x-axis.
Author Response
We thank the reviewer for this comment and now clarified as follows. Our comments find below.

Reviewer 2 Report
In this study, the authors examined the precise molecular mechanisms of carnosine generation and related metabolic actions using Cndp1-knockout (Cndp1-KO) mouse model. This study indicated that a global Cndp1-KO in mice results in a selective and marked, gender- and age-specific increase in renal carnosine and anserine concentrations, in higher renal Hsp70 expression and an altered renal amino acid profile. Blood glucose homeostasis and the response to intraperitoneal insulin is gender specifically modified, body weight gain improved. The experiments reported and the conclusions drawn from them were logical, clear, and well presented. However, I have the following comments.
- It has been suggested that carnosine inhibits the progression and growth of bladder tumor. The authors discuss the relationship between Cndp1 and tumor cells.
- The authors also demonstrated the expression levels of HSP70 in Cndp1-knockout (Cndp1-KO) mouse model. Authors need to discuss the role and function of another HSP family (HSP90, HSPA6….) in Cndp1 gene.
Author Response
Please find our comments attached.

Reviewer 3 Report
The authors describe a novel approach to testing the renal effects of increasing tissue concentrations of carnosine and anserine in mice. The study design is sound and the methods are generally well described. Comments:
The manuscript contains English errors throughout the text; careful editing is warranted.
Title: Suggest change to “A global Cndp1-knock-out selectively increases renal carnosine and anserine concentrations in an age- and gender-specific manner in mice”
Line 26 & throughout: “11-weeks-old, 55-weeks-old” when used as an adjective
Line 28 & elsewhere: carnosine and anserine … 2- to 9-fold, respectively
Line 29: abundantly… but remained unchanged
Line 36: sentence needs reconstruction
Line 52: an ACE
Line 58-9: sentence needs reconstruction
Line 68: metabolite-induced
Line 69: Initial intervention
Line 90: Explain your procedures of tissue collection in detail. Specify the tissue here and throughout.
Line 93: the reaction
Line 98: an additional
Line 115: What time of day was the 5-hour fasting period imposed? During the mouse active or rest period?
Results
Line 240: gender-specific
Line 242: were confirmed
Table 1 and throughout tables, figures: Please show number of mice for female, male, WT and KO groups. Figure 1: “each n=4” means 4 WT male, KO male, WT female and KO female, correct?
What level of statistical power was achieved with n=4 per group? Figure 3 states n=4 and n=14 for age groups – why were these numbers so different?
Line 300: KO and WT mice
Table 2: “In the pancreas…No carnosine-degrading activity…” Should be removed from the table footnote since these data are not shown.
Line 290: synthesis, measured
Line 309: were similar
Line 406: Clarify that the high CN1-activity model was the WT in this study.
Author Response
Please find our comments attached.

Round 2
Reviewer 1 Report
Thanks for providing a detailed response to reviewer´s comments.
Authors provide new information regarding food intake in KO mice [“Food intake was significantly higher in Cndp1-KO as compared to WT mice (p<0.008)”]. Accordingly, the discussion section must be fixed:
i.e. Discussion section: “Since food and energy consumption were not documented, the mechanisms of increased body weight gain remain vague.” Food intake was documented.
i.e. Discussion section: “Blood glucose homeostasis and the response to intraperitoneal insulin is gender specifically modified, body weight gain improved”. Body weight was not improved, actually was worsened.
In the response to reviewers´ letter (question regarding Fig 7) authors state “It is neither explained by food intake (unchanged, data added; all mice received the same diet)”. Why did authors make such statement if they clearly indicate that “Food intake was significantly higher in Cndp1-KO as compared to WT mice (p<0.008)”.
In the response to reviewers´ letter authors performed calculations of AUC during the IP-ITT and showed no significant differences between groups. Therefore, no differences in insulin sensitivity between KO and WT. Why did authors stated in the discussion that insulin sensitivity is gender specifically modified? “Blood glucose homeostasis and the response to intraperitoneal insulin is gender specifically modified, body weight gain improved”
In the revised manuscript, Figures do not appear.
Author Response
Dear Reviewer,
Thanks for your helpful suggestions and please find enclosed the Point-by-Point reponse.
Best wishes on behalf of all authors
Verena

Reviewer 3 Report
The authors have made significant improvements to the manuscript and have addressed reviewers' concerns.
Recommendations:
Use "11-week-old" or "55-week-old" consistently; the authors use both "weeks-old" and "week-old." "Weeks" in figure legends needs correction.
Food Intake description: "Food intake was determined by measuring the decrease in food cup weight..." rather than "decrease in nutrition..."
Include in the discussion the small sample sizes, which is a limitation that increases the risk of making a type-II error.
Author Response
Dear Reviewer,
Thanks for your helpful suggestions and please find enclosed the Point-by-Point reponse.
Best wishes on behalf of all authors
Verena Peters

Round 3
Reviewer 1 Report
No further comments